# A Gas Turbine Cooled-Stage Expansion Model for the Simulation of Blade Cooling Effects on Cycle Performance

**Roberta Masci and Enrico Sciubba \***

Department of Mechanical and Aerospace Engineering, University of Roma Sapienza, 00185 Roma, Italy;
roberta.masci1@gmail.com
**\*** Correspondence: enrico.sciubba@uniroma1.it

**Abstract:** Modern gas turbine firing temperatures (1500–2000 K) are well beyond the maximum allowable blade material temperatures. Continuous safe operation is made possible by cooling the HP turbine first stages, nozzle vanes and rotor blades, with a portion of the compressor discharge air, a practice that induces a penalty on the thermal efficiency cycle. Therefore, a current issue is to investigate the real advantage, technical and economical, of raising maximum temperatures much further beyond current values. In this paper, process simulations of a gas turbine are performed to assess HP turbine first-stage cooling effects on cycle performance. A new simplified and properly streamlined model is proposed for the non-adiabatic expansion of the hot gas mixed with the cooling air within the blade passage, which allows for a comparison of several cycle configurations at different turbine inlet temperatures (TIT) and total turbine expansion ratio (PR) with a realistically acceptable degree of approximation. The calculations suggest that, at a given PR, the TIT can be increased in order to reach a higher cycle efficiency up to a limit imposed by the required amount and temperature of the cooling air. Beyond this limit, no significant gains in thermal efficiency are obtained by adopting higher PR and/or increasing the TIT, so that it is convenient in terms of cycle performance to design at a lower rather than higher PR. The small penalty on cycle efficiency is compensated by the lower plant cost. The results of our model agree with those of some previous and much more complex and computationally expensive studies, so that the novelty of this paper lies in the original method adopted on which the proposed model is based, and in the fast, accurate, and low resource intensity of the corresponding numerical procedure, all advantages that can be crucial for industry needs. The presented analysis is purely thermodynamic and it includes no investigation on the effects of the different configurations on plant costs. Therefore, performing a thermo-economic analysis of the air-cooled gas turbine power plant is the next logical step.

**Keywords:** blade cooling; gas turbine efficiency; TIT-expansion ratio for optimal efficiency

---

## 1. Introduction

Turbine inlet temperatures (TIT in the following) of 1500–2000 K have become a standard for most modern advanced gas turbine applications, so that both the stationary vanes and rotating blades of the first stage (and often of the second and third as well) need to be properly cooled [1–3]. The cooling medium is usually a small portion of the HP compressor discharge air, which represents a direct loss in terms of engine power output and thermal efficiency. Notice that the TIT used throughout this paper is the maximum allowable turbine inlet temperature, and is different from the TIT-ISO, which is calculated as the temperature resulting from mixing all cooling flows and the combustor exit flow in a single point. It is also obviously lower than the combustor outlet temperature -COT- sometimes used

in similar studies. Therefore, the gains in thermal efficiency with increasing TIT must be significant enough to justify the added complexity and cost of the cooling system. In this work, focus is placed on the proper assessment of the real advantage of raising maximum cycle temperatures significantly beyond current values. In general, further TIT increases are nullified by the inability of materials to cope with such high temperatures, so that new casting technologies and costlier superalloys come into play. Therefore, a cost balance must be made in order to establish whether the gain in thermal efficiency brought about by an increase in the TIT outbalances the cost of using more expensive high-temperature, creep-resistant alloys. Since in an economic analysis the increase in installation and maintenance cost is compensated by a higher fuel efficiency, a thermodynamic cycle analysis is the first necessary step in the process. Due to the large number of parameters influencing cooling losses [4–6], accurate cycle calculations can only be attempted for specified gas turbines models and operating conditions [2,7]. Satisfactory predictions are thus provided only for the performance of a particular configuration. In order to obtain a general understanding, approximate and simpler models are needed. However, an obvious drawback of such models is that, if oversimplified, they cannot provide satisfactory numerical results. Several models are synthetically reviewed by Horlock et al. [8]. They identify two different approaches, whose common feature is that of considering losses in stagnation temperature and pressure as the major result of the mixing of cooling air with the hot gas main flow. For the purpose of the present study, it is important to identify the different features of such approaches:

(1) *Continuous heat and work extraction*: This model was firstly presented by El-Masri [9], then modified first by De Paepe and Dick [10] and later revisited by Bolland and Stadaas [11]. It models the turbine stage as an expansion path with continuous work extraction. The gas stagnation temperature is approximated by a continuously varying function, rather than the actual stepwise variation. In this way, the assumed temperature profile underestimates the relative gas-to-surface temperature difference in the stator and correspondingly overestimates that in the rotor; however, the stage average is approximately preserved. This rather strong assumption was first proposed by El-Masri to obtain closed-form solutions for the states along the expansion path. Then, De Paepe and Dick adapted such model to a cycle analysis of steam-injected gas turbines, proposing a new representation of the cooled expansion process, depicted in Figure 1. They divide the global pressure drop into a large number of intervals, each of them representing a cooled turbine stage, and each of them represented as a succession of:

(1)     Adiabatic expansion (work extraction);
(2)     Mixing of cooling fluid and hot gas at constant pressure (loss of stagnation temperature);
(3)     Mixing of cooling fluid and hot gas at constant temperature (loss of total pressure).

(2) *Simple stage-by-stage model*: This model was developed by Jordal [12] and adopted by Horlock et al. [8] in their study about the limitations on gas turbine performance induced by excessive blade cooling flows. It considers the effect of cooling on a discrete series of individual blade rows and treats cooling and expansion separately. In this model the stator and rotor cooling and the consequent mixing of hot gas and coolant are considered to take place immediately downstream of the respective expansions.

The targets of the two presented approaches are definitely different. The final aim of the first one is that of achieving closed-form solutions, which may be applied to a large number of operating conditions and parameters. In the second one, the focus is placed on obtaining the most realistically accurate results for a specific case. Jordal et al. [13], in their study on the behavior of oxyfuel cycles operating at high combustor outlet temperatures, presented three models for turbine blade cooling, one belonging to the first group, and two belonging to the second group. Walsh and Fletcher [14] presented a third model in which the maximum cycle temperature is the outlet temperature at the discharge of the nozzle guide vanes (NGV), $T_{g,so}$, instead of the TIT. Under this assumption they calculated the amount of cooling air mass flow rate by means of semi-empirical correlations or diagrams for given values of $T_{g,so}$. Sanaye and Darvishi [15] reviewed the three models and compared them by estimating, for each of them, the amounts of cooling air required by sixteen gas turbines in four ranges of power outputs. They concluded that, while all three models provide an accuracy better than

3% for the estimated power output, thermal efficiency, and turbine outlet temperature, they differ as to their applicability to different turbine "power classes". Specifically, the De Paepe model provides the best agreement with the experimental values of the turbine outlet temperature and of the thermal efficiency $\eta_{th}$. The Jordal model provides the best estimates for the net work output $W_{net}$ for turbines up to 80 MW and the Walsh-Fletcher model for those between 80 and 180 MW, but their performance reverses when it comes to the other classes (Jordal is worst between 80 and 180 and Walsh-Fletcher worst below 80 MW).

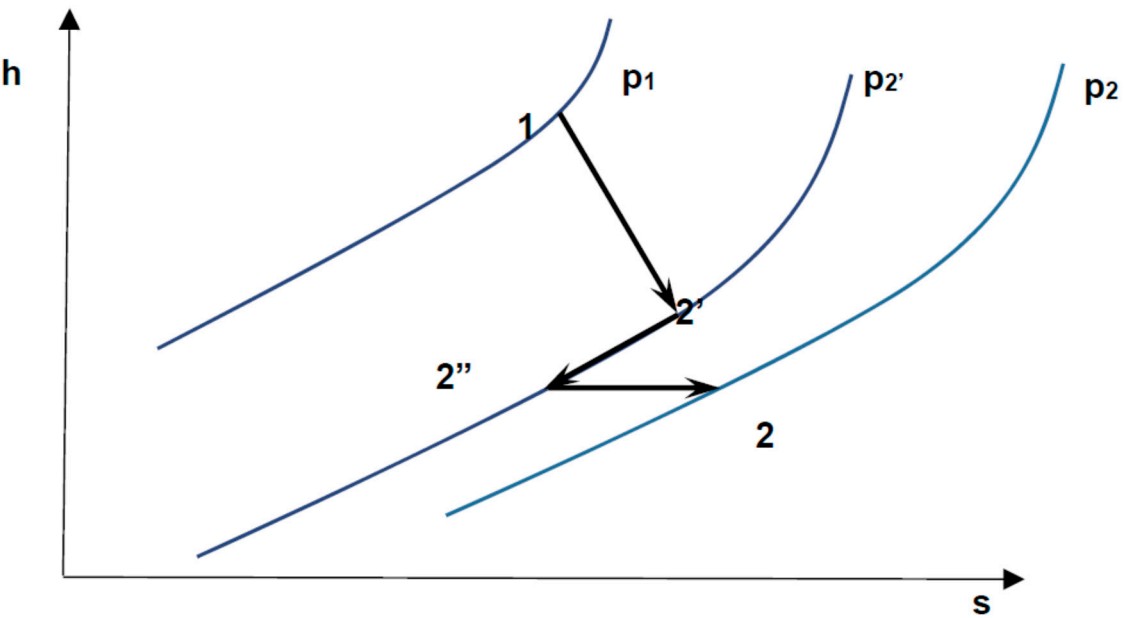

**Figure 1.** De Paepe's cooled expansion with mixing losses (adapted from [10]).

## 2. The Proposed Model for Cooled Expansion

In this paper, a novel model for cooled stage expansion is proposed. The model allows for sufficiently accurate, but not extremely resource-intensive, calculations of different cycle configurations, while retaining adequate detail to provide results of practical engineering interest. The model is a synthesis of the two above-mentioned approaches. It was decided to apply the De Paepe cooled expansion model separately to the stator and to the rotor row, thus considering a discrete (i.e., only in the rotor), rather than continuous, work extraction. In this way, the mixing of hot gas and coolant takes place first downstream of the stator row and afterwards downstream of the rotor row. A representation of such an expansion process is given in Figure 2, where the line 1–2′ represents the cooled expansion without work extraction, and 2′–2″ and 2″–2 respectively the temperature and pressure drops due to mixing in the NGV row, while 2–3′ represents the cooled expansion with work extraction, followed by the temperature (3′–3″) and pressure losses (3″–3) due to mixing in the rotor row.

Since the gas temperature profile is known along the cooled expansion both in the stator and rotor row, the gas temperatures $T_{2'}$ and $T_{3'}$ at the end of each cooled expansion are pre-assigned.

The pressures $p_{2'}$ and $p_{3'}$ are also known once the pressure ratio and degree of reaction of the turbine are prescribed. Therefore, the final temperature and pressure of the gas after both mixings ($T_2$, $p_2$, $T_3$, $p_3$) are the only unknowns.

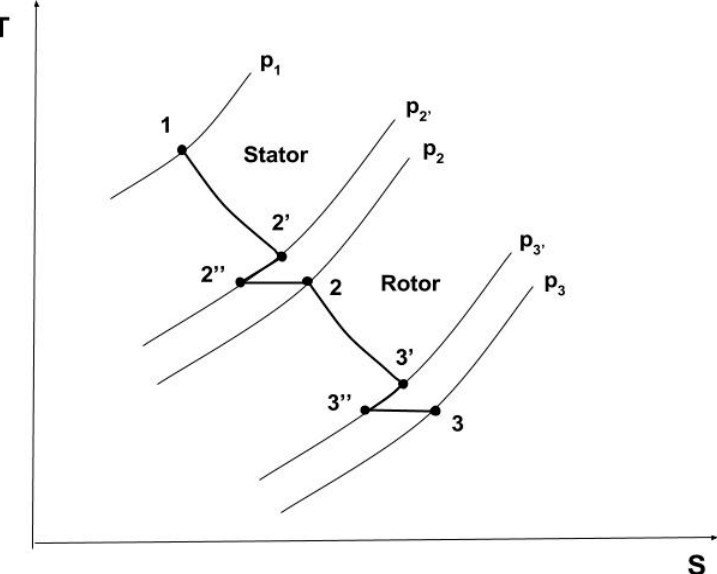

**Figure 2.** The proposed cooled expansion.

The temperature $T_2$ can be obtained from the stator global energy balance (Notice that all temperatures considered here are static temperatures: For the sake of simplicity, the effect of different velocity values across successive statoric passages have been neglected, though they can easily be included in the calculation),

$$T_2 = \frac{m_g c_{p,g} T_1 + m_{c,NGV} c_{p,c} T_c}{m_{out,NGV} c_{p,mix}} \tag{1}$$

where $c_{p,g}$, $c_{p,c}$ and $c_{p,mix}$ are the specific heats [J/kgK] of the gas, coolant, and mixed fluid respectively, $T_1$ is the gas inlet temperature [K], $T_c$ is the coolant temperature [K], and $m_{out,NGV}$ is the stator outlet mass flow rate [kg/s], derived from the mass balance of the mixture,

$$m_{out,NGV} = m_g + m_{c,NGV} \tag{2}$$

where $m_g$ and $m_{c,NGV}$ are the gas and the NGV coolant mass flow rates respectively (the latter derived from blade cooling calculations). Assuming that the mixing reduces the momentum of the hot gas, because the coolant flow emerging from the blade surface must be turned and accelerated to the direction and speed of the main flow, such momentum reduction can be expressed as a pressure loss for the main hot gas flow. If $v$ is the coolant average velocity at the mixing point, the momentum reduction of the main flow per unit time is $vm_c$, while the corresponding pressure loss is $dpA$, $A$ being the flow area. The momentum balance of the mixing is hence:

$$vm_c = -dpA \tag{3}$$

The flow area $A$ is derived from the gas mass flow rate relation:

$$A = \frac{m_g}{\rho v} \tag{4}$$

and substituted in Equation (3), obtaining the following expression for the momentum balance:

$$-dp = \frac{m_c}{m_g} \rho v^2 \tag{5}$$

Or:

$$-\frac{dp}{p} = \frac{m_c}{m_g}\frac{\rho v^2}{p} \tag{6}$$

Assuming an ideal behavior for the hot gas, from the ideal gas law $p/\rho = RT$ and the Mach number definition:

$$Ma = \frac{v}{\sqrt{\gamma RT}} \tag{7}$$

We finally obtain the following expression for the pressure losses in the mixing:

$$-\frac{dp}{p} = \frac{m_c}{m_g}\gamma Ma^2 \tag{8}$$

Therefore, the pressure loss ($p_{2'}-p_2$) and consequently the pressure $p_2$, can be derived from the momentum balance of the stator mixing [10] (There are of course additional pressure losses due to leading edge showerhead cooling, leakage, frictional losses in the internal cooling channels etc. These are all proportional to the square of the coolant mass flowrate, and are taken into account automatically by the simulation code via standard loss coefficients):

$$\frac{p_{2'} - p_2}{p_2} = \frac{m_{c,NGV}}{m_g}\gamma Ma^2 = \frac{m_{c,NGV}}{m_g}\xi \tag{9}$$

where $\gamma = (k-1)/k$ is the adiabatic exponent ($k$ being the heat capacity ratio) and $Ma$ is the Mach number of the flow immediately upstream of the mixing. Since mixing happens in the aft section of the blade (near the trailing edge), $Ma$ is near unity [10], and assuming $Ma = 0.8$ it follows $\gamma Ma^2 = 0.9$. Applying the same procedure to the rotor row, $T_3$ can be derived from the rotor global energy balance:

$$T_3 = \frac{m_{out,NGV}c_{p,mix}T_2 + m_{c,rot}c_{p,c}T_c - P}{m_{out,rot}c_{p,mix}} \tag{10}$$

where $m_{out,rot}$ is the rotor outlet mass flow rate, given by the mass balance:

$$m_{out,rot} = m_{out,NGV} + m_{c,rot} \tag{11}$$

While $P$ is the turbine stage power output, which is known for the selected turbine stage and is given by:

$$P = m_g\psi U^2 \tag{12}$$

$\psi$ being the stage loading coefficient (known for a given cascade). The pressure loss ($p_{3'}-p_3$), from which $p_3$ can consequently be computed, is estimated from Equation (7):

$$\frac{p_{3'} - p_3}{p_3} = \frac{m_{c,rot}}{m_g}\gamma Ma^2 = \frac{m_{c,rot}}{m_g}\xi \tag{13}$$

The pressure losses can be finally translated into a reduction of the turbine polytropic efficiency $\eta_{p,t}$. Therefore, a new cooled turbine reduced-polytropic efficiency $\eta_{p,t\,cool}$ can be defined, whose value is given by an expression first proposed by Jordal & Bolland [12]:

$$\eta_{p,t,c} = \eta_{p,t} - \left[\ln\beta\frac{1}{\left(1-\frac{1}{\beta}\right)}\frac{m_c}{m_g}S\right] \tag{14}$$

where $\beta$ is the net stage expansion ratio, $m_c$ the total coolant mass flow rate, given by:

$$m_c = m_{c,NGV} + m_{c,rot} \tag{15}$$

*S* is a parameter which depends on the engine technology level that takes lower values for large and/or modern engines and higher values for old and/or small engines. According to Jordal, a typical state-of-the-art value for *S* is 0.1 [12]. This parameter was also reintroduced in [13] via a semi-empirical reasoning, as the result of a data-fitting of the ratio of the efficiency of the cooled cycle to that of the uncooled one for turbines of different years (i.e., state-of-the art of the cooling technology). While it is clear that *S* somehow lumps the viscous/thermal losses in the cooling system, there is no rigorous derivation for it, and it ought to be considered a semi-empirical adjustment.

From the $\eta_{p,t\,cool}$ value, a new efficiency for the cooled stage can be also derived from the relation,

$$\eta_{stage} = \frac{\beta_{rot}^{(\gamma\,\eta_{p,t,c})} - 1}{\beta_{rot}^{\gamma} - 1} \tag{16}$$

where $\beta_{rot}$ is the rotor expansion ratio.

Summarizing, the originality of the proposed model lies in the flowchart of its calculations:

(a)　The uncooled statoric expansion is calculated by a process simulator (process 1-2′);
(b)　Then, the mass flowrate of cooling air necessary to maintain the assigned $T_{stator\,max}$ is calculated;
(c)　An energy balance provides the temperature of the mixture (point 2″);
(d)　A mode equation provides then the gas temperature and pressure at rotor inlet (point 2);
(e)　The process simulator is again used to calculate the uncooled expansion of the $m_2$ mixture;
(f)　Points b,c,d are repeated for the rotor;
(g)　The process simulator is run once more with the corrected values of mass flowrates, to calculate the engine performance.

## 3. Air-Cooled GT Power Plant Simulations

The above model was implemented in the in-house process simulator CAMEL-Pro$^{\text{TM}}$ in order to perform a thermodynamic cycle analysis of a gas turbine power plant (Figure 3). Since only the effects of turbine first-stage cooling on gas turbine performance are analyzed in this study, the turbine was divided into a HP cooled stage topping a series of uncooled ones. Different sets of cycle simulations were performed to calculate the cycle efficiency defined as,

$$\eta_{cycle} = \frac{\alpha\Delta w_{cycle}}{LHV} = \frac{\eta_{mech}\,\alpha(w_T - w_C)}{LHV} \tag{17}$$

where $w_T$ (which includes the factor $1 + 1/\alpha$) and $w_C$ are respectively the specific work produced by the turbine and absorbed by the compressor, so that their difference ($w_T - w_C$) represents the gas turbine net specific work output and $\eta_{mech}$ is the mechanical efficiency. The air-to-fuel ratio is defined as $\alpha = \frac{(LHV + h_f)\eta_c - c_{p,TIT}T_{TIT}}{c_{p,c}T_c - c_{p,TIT}T_{TIT}}$, where $T_c$ is the compressor outlet temperature, *LHV* is the lower heating value of the fuel and $\eta_c$ the combustion efficiency (in the CAMEL-Pro simulator used here, the specific heats are calculated as functions of the local temperature).

The design parameters listed in Table 1 were extracted and/or adapted from technical documents in the public domain, and were kept constant in all simulations. The compressor inlet mass flow rate was set equal to 30 kg/s, and methane $CH_4$ was assumed as fuel, with a *LHV* = 50,000 kJ/kg.

Before proceeding with the power plant simulations, which evaluate cycle performance in terms of net specific work and thermal efficiency, a model validation was performed by comparing the uncooled and the air-cooled gas turbine cycle.

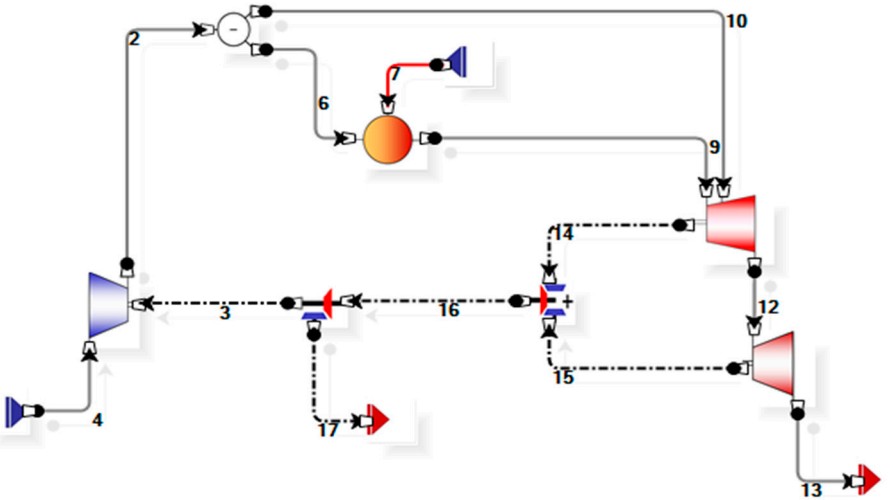

**Figure 3.** Schematic representation of the gas turbine in CAMEL-Pro$^{TM}$ simulator.

**Table 1.** Design parameters values kept constant throughout all simulations.

| Parameter | Compressor | Turbine | Combustion Chamber |
|---|---|---|---|
| $\eta_p$ | 0.85 | 0.88 | - |
| $\eta_{ad}$ | 0.79 | 0.83 | - |
| $\eta_{mech}$ | 0.98 | 0.98 | - |
| $\eta_{cc}$ | - | - | 0.98 |
| $\Delta p\%$ | - | - | 2 |

*Model Validation*

The first set of simulations was performed to compare the uncooled with the air-cooled cycle. For a PR = 30, coolant mass flow rates that guarantee a maximum blade temperature equal to 1230 K were calculated for each value of the TIT with a MATHEMATICA code developed by the authors [16–18], setting the coolant temperature equal to the compressor discharge temperature. The coolant mass flow rates increased with TIT in both stator and rotor with a slightly sub-linear trend, as depicted in Figure 4.

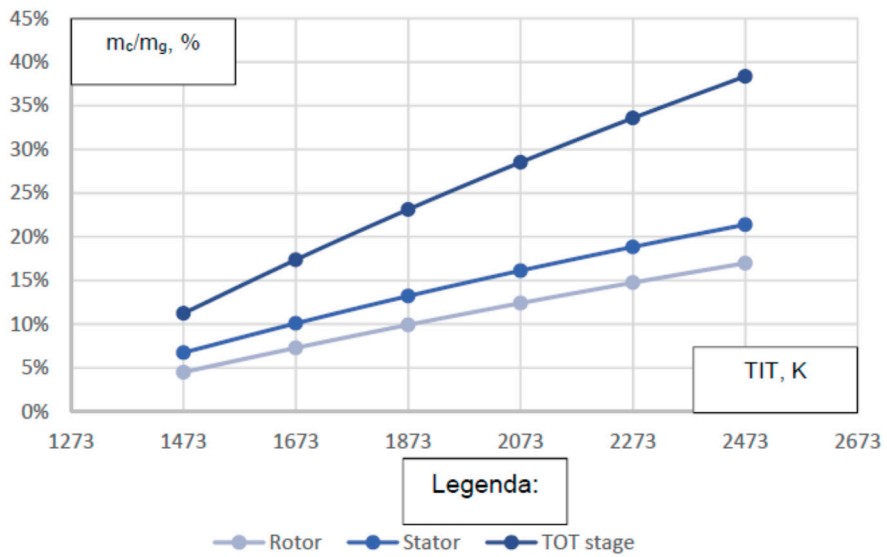

**Figure 4.** Coolant mass flow rates ((total turbine expansion ratio (PR) = 30).

The relatively low efficiency shown in Figure 4 is typical of the simple, non-regenerated cycle assumed here. As expected, both power output and efficiency values of the uncooled cycle are slightly higher than those of the air-cooled cycle (Figure 5), and the relative difference increases with TIT. However, since the uncooled cycle efficiency trend is an ideal one, such high values of TIT exceed the creep limits on the blade material, and hence could not be attained without cooling. Therefore, only if the penalty introduced on the thermal efficiency by first-stage cooling are outweighed by sufficiently higher TITs the efficiency gain is significant enough to justify the added complexity and cost of the cooling system.

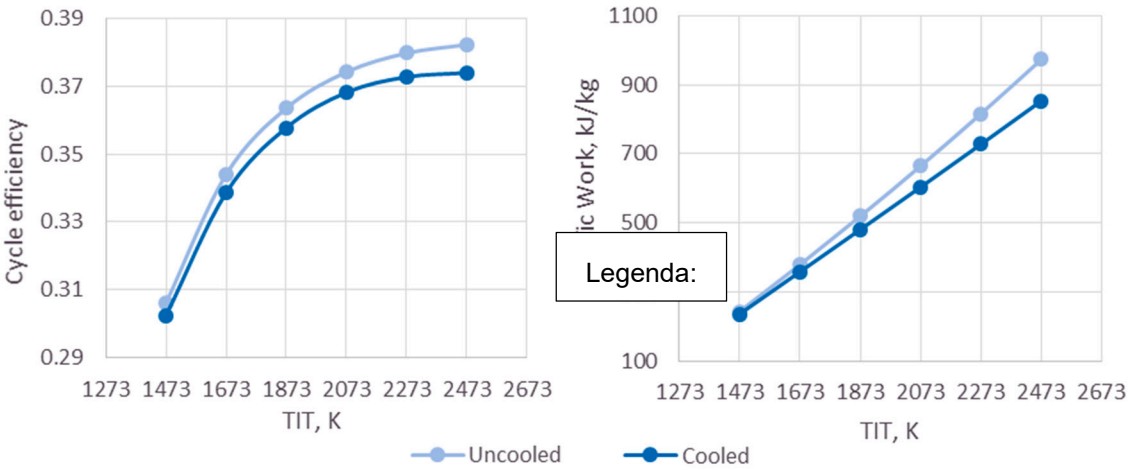

**Figure 5.** Cycle efficiency and specific work output as a function of turbine inlet temperatures (TIT) (PR = 30).

A very similar trend to that shown in Figure 5 can be found in Horlock's study about limitations on gas turbine performance imposed by large turbine cooling flows [8].

In that paper, the authors analyzed the effect of three-stage cooling on cycle efficiency, demonstrating that it leads to a more substantial drop than that caused by first-stage cooling only. The proposed model provides results in accordance with that investigation, and can therefore be considered validated, at least under this respect.

## 4. Results and Discussion

A set of cycle simulations was performed varying both pressure ratios ($\beta_{max}$ from 10 to 40) and turbine inlet temperatures (*TIT* = 1473–2673 K) in order to analyze the air-cooled gas turbine cycle performance. Inlet cycle pressure and temperature were kept constant throughout ($p_0 = 1$ bar, $T_0 = 300$ K). The values of the required cooling flow rates are shown in Figure 6a.

Let us first analyze the effect of different coolant flowrates on the air-to-fuel ratio $\alpha$. For a given *PR*, $\alpha$ decreases with increasing TIT, as shown in Figure 6b, for two reasons:

(1)　In order to reach higher temperatures, more fuel must be injected in the combustion chamber;
(2)　A higher air extraction rate from the HP compressor stages for cooling purposes, corresponds to a decrease of the air flowrate into the combustion chamber.

The resulting trends of $\alpha$ as a function of the amount of coolant used for each combination of PR and TIT are shown in Figure 7.

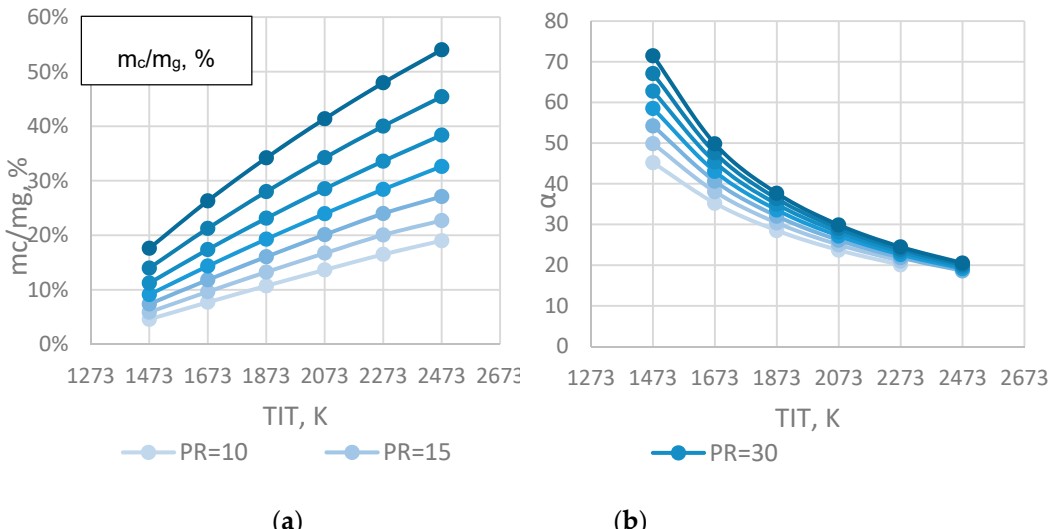

**Figure 6.** Cooling mass flow rates vs. (**a**) air to fuel ratio (**b**) TIT, for different PR.

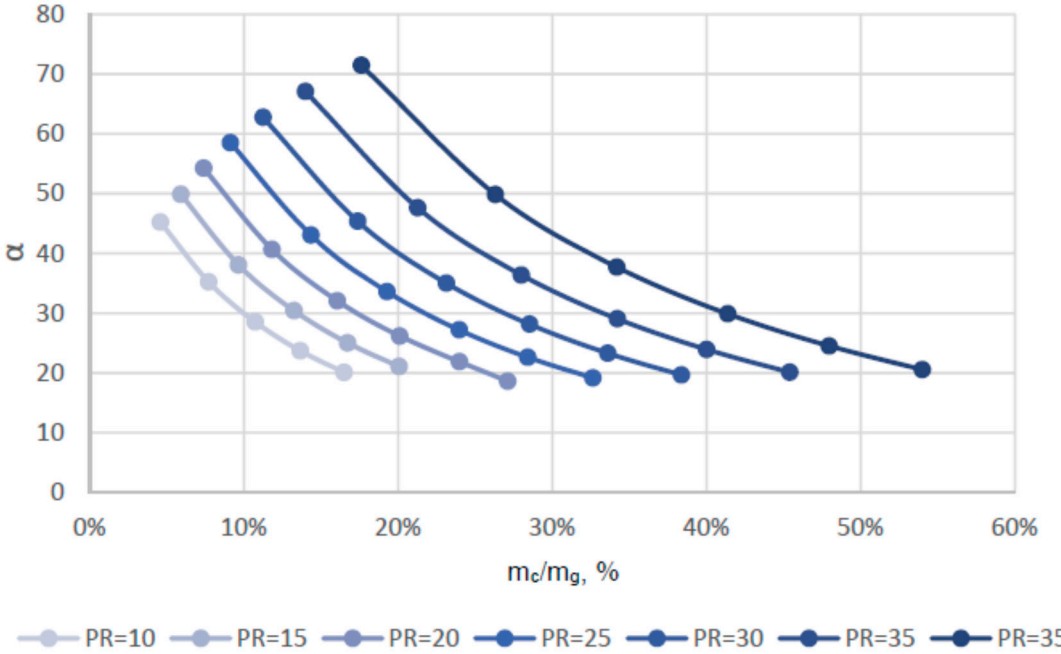

**Figure 7.** Air-to-fuel ratio vs. cooling mass flow rates for increasing PR.

Since the compressor inlet mass flow rate and pressure ratio are kept constant in this set of simulations, the reduction in net power output is caused by the decrease in turbine work output brought about by larger amounts of cooling air. The turbine specific work output is given by the sum of the cooled first stage and uncooled downstream stages work outputs, the latter representing the major portion of the whole. As the TIT increases, so does the cooling mass flow rate, and the portion that participates in the combustion and produces work in the turbine is hence reduced. On the other hand, $T_c$ (assumed here to be equal to the compressor discharge temperature) increases with the cycle pressure ratio PR, leading to a (small) decrease in the air-to-fuel ratio $\alpha$ and to a correspondingly small increase in $\eta_{cycle}$.

Our calculations confirm that the two phenomena do not balance out, so that the discharge temperature from the first turbine stage resulting from the mixing of the hot gas and the coolant drops

as well, and less work is produced leading to a lower available enthalpy drop in the downstream uncooled turbine stages.

Confirming a well-known trend [3,19,20], for each value of TIT there is an optimal pressure ratio that guarantees maximum cycle efficiency, but not maximum work output (see Figure 8c,d). Such efficiency maximum moves towards higher pressure ratios as the TIT increases. This also reproduces the well-known trend observed in real processes, and constitutes a further validation of our model.

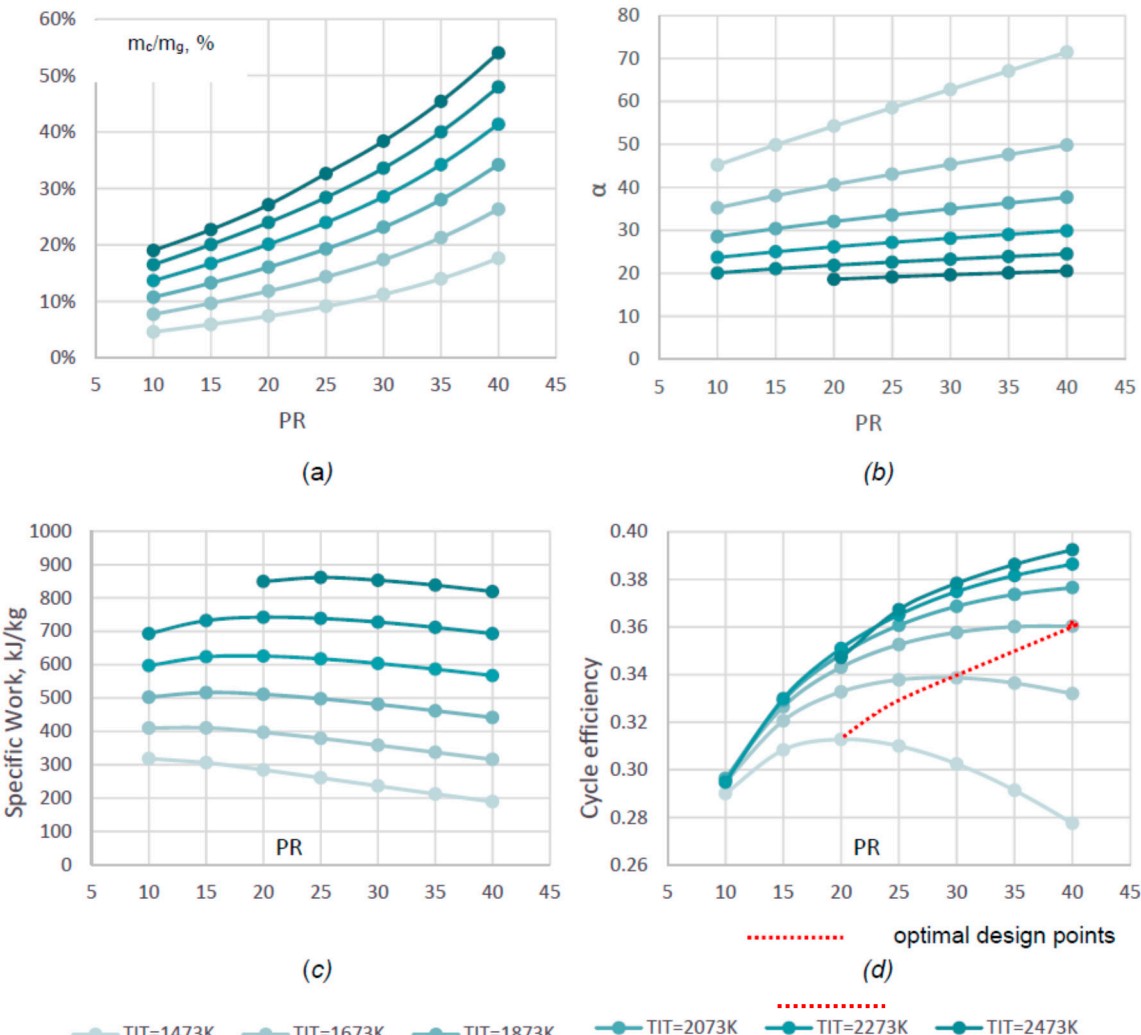

**Figure 8.** Cooling mass flow rates (**a**) air to fuel ratio, (**b**) specific work and, (**c**) cycle efficiency versus (**d**) TIT for increasing $\beta_{\text{max}}$.

The gains in thermal efficiency and fuel savings brought about by higher TITs can hence be obtained only with the penalty of an increased compressor cost (with possibly one or two additional stages or with a different stage design). Figure 8a suggests that for each pressure ratio there is a limiting combustion temperature (well below the one set by stoichiometric combustion) beyond which cycle thermal efficiency decreases. For example, at PR = 20, raising the TIT beyond 2273 K does not bring any improvement in cycle efficiency, on the contrary, there is a drop in efficiency for TITs exceeding this value.

As for the specific work, Figure 8b shows that, for each value of the TIT, it decreases with higher pressure ratios. Though not detectable in the graphs presented here, this decrease is higher than the corresponding decrease observed for constant coolant mass flow rate. This is explained by the

fact that, as the compressor discharge temperature approaches the allowable blade temperature, the cooling requirements become so large that the first-stage outlet temperature drops (because of the mixing of the hot gas with a large amount of "cold" air) and the turbine work is consequently reduced. Furthermore, the increase of the required cooling mass flow rate with the TIT leads to substantially increased stagnation pressure losses that induce a corresponding drop in the expansion ratio across the turbine, annihilating the advantage of a higher TIT.

Finally, since the specific work increases with the TIT, the requirement for the maximum possible power output is a valid reason to increase the TIT, which may outweigh efficiency consideration. Cycle simulations with constant net power output were also performed, and similar cycle efficiency trends were found.

## 5. Conclusions

The calculations presented in this paper show that, in almost complete analogy with uncooled gas turbine cycles, the thermal efficiency of an air-cooled gas turbine at a certain technological level is primarily dependent on the turbine inlet temperature TIT and on the pressure ratio PR. At a given pressure ratio, a maximum in thermal efficiency is likely to be reached below stoichiometric conditions. There is a point though at which the large quantity of cooling air required for such high TITs induces excessive thermal and pressure losses that reduce the enthalpy drop in the turbine, decreasing the specific work output and lowering the thermal efficiency (the fuel consumption remains almost the same). Therefore, gains in cycle efficiency with increasing TIT up to the stoichiometric limit become marginal if not negligible, while the benefit of higher power outputs is always attained, although mixing losses reduce the corresponding benefits. As the pressure ratio increases, the upper temperature limit increases as well, and higher cycle efficiencies are reached.

These findings suggest that, at a given PR, gas turbine designers should search for a suitable compromise between increased TIT and cycle efficiency. Moreover, when choosing the optimal *PR*, they should decide whether to accept a higher initial cost due to a larger compressor (higher *PR*) but with future fuel savings (higher efficiency), or a lower efficiency with lower installation costs (lower PR).

It is concluded that, for a given gas turbine technology level, turbine inlet temperatures can be increased in order to reach higher cycle efficiency, but there actually exists a limit imposed by the required amount of cooling air. In order to take advantage of further TIT increases, improvements in cooling technology are certainly necessary, to attain higher cooling effectiveness, so that smaller amounts of cooling air will be needed for the same TIT to maintain an acceptable metal temperature. Emphasis must also be placed on raising the allowable blade material temperature by advances in metallurgical technology. The analysis presented in this paper is purely thermodynamic, with no investigation of the effects of the modified system configurations on plant costs, such that future work addressing a thermo-economic analysis of the air-cooled gas turbine power plant is the next step required to bring the results into the industrial domain.

**Author Contributions:** E.S. conceived the model and the thermodynamic procedure; R.M. derived the formulae and performed the calculations. Both contributed equally to the writing of the paper.

**Funding:** RM research was partially funded by a grant by GA Industries, Roma

**Conflicts of Interest:** The authors declare no conflict of interest.

## Nomenclature

| | |
|---|---|
| *A* | Passage flow area [m$^2$] |
| $c_p$ | Specific heat [J/(kg K)] |
| *COT* | Combustor Outlet Temperature [K] |
| *LHV* | Lower Heating Value [kJ/kg] |
| *m* | Mass flow rate [kg/s] |
| *Ma* | Mach number |

| | |
|---|---|
| *NGV* | Nozzle Guide Vane |
| *p* | Pressure [Pa] |
| *P* | Power [W] |
| *PR* | Cycle pressure ratio |
| *R* | Gas constant [kJ/(kgK)] |
| *S* | Efficiency penalty parameter |
| *T* | Temperature [K] |
| *TIT* | Turbine Inlet Temperature [K] |
| *TOT* | Turbine Outlet Temperature [K] |
| *v* | Circumferential velocity [m/s] |
| *w* | Specific work [J/kg] |

**Greek Symbols**

| | |
|---|---|
| $\alpha$ | Air-to-fuel ratio |
| $\beta$ | Turbine expansion ratio |
| $\Delta p\%$ | Pressure loss [%] |
| $\eta$ | Efficiency |
| $\gamma$ | Polytropic exponent |
| $k$ | Isentropic exponent, $c_p/c_v$ |
| $\rho$ | Density [kg/m$^3$] |
| $\psi$ | Stage loading coefficient |

**Subscripts**

| | |
|---|---|
| 1,2,3 | Referring to state 1,2,3 |
| ad | Adiabatic |
| c | Coolant |
| C | Compressor |
| CC | Combustion Chamber |
| g | Gas |
| in | Inlet |
| mech | Mechanical |
| mix | Mixing |
| NGV | Nozzle Guide Vane |
| out | Outlet |
| p | Polytropic |
| rot | Rotor |
| so | Stator Outlet |
| st | Stage |
| T | Turbine |
| th | Referred to cycle |

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
