# Peer review of "A Gas Turbine Cooled-Stage Expansion Model for the Simulation of Blade Cooling Effects on Cycle Performance"

_ijtpp, doi:10.3390/ijtpp4040036_

Round 1
Reviewer 1 Report
Please find below my specific comments for the paper along with more general comments. In essence, further elaboration of the novelty of the paper would be helpful to the reader. Comparison to the existing methods along with any time benefit using this method is really needed to fully gauge the potential benefit of the method being proposed by the authors.
Figure 1 is referenced in line 73; however, the figure is not provided in the manuscript. The authors comment on Sanaye’s reviewing of the three models presented in the literature review but not on any conclusions from that study which I feel would be helpful in framing the current work performed by the authors. Line 103 mentions two approaches which were discussed in the literature review although a third was briefly mentioned. I’m not sure the rearranged version of equation 4 if really necessary to add. The parameter S discussed in lines 166 - 168 needs some sort of physical interpretation – it otherwise appears to be just a random factor without some sort of physical context. References for the values if table 1 would be useful (even if some are just average, generic values). Reference 14 is cited with regards to the MATHEMATICA code (lines 221 - 222) – however, this does not seem to be available online being an in-house master’s thesis. Therefore, this should be added in an appendix here if the reference cannot be accessed. The authors discuss what they term the fuel-to-air ratio which they also define. However, fuel-to-air ratio implies and consequently is confusing when looking at figures 6 - 8. As the authors discuss, the mass flow of fuel to air is likely to increase with turbine inlet temperatures. To avoid confusion, the term should either read air-to-fuel ratio or be termed something else. The authors suggest that the 2 main models discussed in the introduction may have a more limited use than that being presented. However, a direct comparison is required with the results of the proposed model to fully ascertain the benefits of the proposed model/limitations of the cited models. It would also be helpful to display standard thermodynamic cycle lines (for pressure ratio, turbine inlet temperature and cycle efficiency) for an uncooled blade.
General Comments:
The novelty of the method presented in the paper over alternate methods needs further elaboration There should be a space between the SI units and the number (e.g. 2000 K) When a range of numbers is given (e.g. 1500-2000), the text is clearer with a space (e.g. 1500 - 2000) Not all symbols and abbreviations appear in the nomenclature and should be added. This includes subscripts All symbols in the main text should be italicised There are quite a number of formatting issues and missing symbols – some perhaps happened when the document was saved as a PDF. Clearly, this will need to be rectified. There are also sometimes variations in font size and spacing (see for example line 117), line spacings (see for example lines 169 and 170) etc. Additionally, I would suggest an empty line above and below each equation. Equation numbering also appears in random places and sometimes overlaps – again perhaps just an issue in the conversion to PDF References should be arranged in order of numerical appearance rather than alphabetically. Mass flow should appear as and not simply m. I would suggest replacing the symbol µ used here to avoid confusion with dynamic viscosity. Particularly in graphics, symbols are not always subscripted (e.g. mc/mg in Fig 4 should be etc.) TOT in figure 4 is not defined in nomenclature Would suggest use of commas for some symbol subscripts, for example cp,g, cp,c, and cp,mix in line 119 In general figure quality could be improved (reducing line thickness in places as lines become indistinguishable sometimes, increasing resolution etc.)

Author Response
rebuttal attached

Reviewer 2 Report
Improve pictures:
highlight the optimum, if any. For example Fig * (d) show us the optimum of Cycle efficiency as a function of PR and TIT. You can connect this point and preset the trend.
Fig 3 include numbering, but it is not in the text = little confusing for reader. Reference to figure 3 on the text??
Fig 2, scheme, I understand this is just the scheme, but, isobars on T-S diagram are definitely not parallel lines as you show on the Fig. 2.
Greek, page 12 :
"dp%" is not Greek. Correct all mistakes on your text!
Nondimensionality: you combine both, dimensioned and nondimensioned parameters. I agree the specific TIT is important for specific application, but can be use T3/T1 similary as you are using epsilon (p2/p1). It is not critical issue, but think about generalise your results.
also
Author Response
rebuttal attached

Round 2
Reviewer 1 Report
Please find below any additional comments from my paper review:
The quality of some of the figures (mainly those reproduced) need to be improved or re-drawn. At the very least the labels should be replaced for legibility. For example, the labels in Figure 1 (such as the x-axis label), are nearly illegible. I think it should still be possible to subscript mc and mg in the graphic software used (I think Excel?). Ideally this should be done If a couple of sources could still be added (in addition to the added sentence) for the figures in Table 1, that would be useful (for example Jet Propulsion by Cumpsty quotes similar figures) Figure 6 comes after Figure 7 in the copy I have receivedAuthor Response
Figure 1 redrawn
Captions mc & mg adjusted
figres 6&7 numbering corrected
Reviewer 2 Report
Satisfied with comments and changes.
Author Response
Thanks.